# Modulation of the Intraseasonal Variability of Early Summer Precipitation in Eastern China by the Quasi-Biennial Oscillation and the Madden-Julian Oscillation

Zefan Ju[1], Jian Rao[1*], Yue Wang[1], Junfeng Yang[2], and Qian Lu[1]

[1]Collaborative Innovation Center on Forecast and Evaluation of Meteorological Disasters / Key Laboratory of Meteorological Disaster of Ministry of Education, Nanjing University of Information Science and Technology, Nanjing 210044, China
[2]National Space Science Center, Chinese Academy of Sciences, Beijing 100190, China

*Correspondence to*: Dr. Jian Rao (raojian@nuist.edu.cn)

**Abstract.** Using the reanalysis and multiple observations, the possible impact of the Madden-Julian Oscillation (MJO) on early summer (June-July) rainfall in eastern China and its modulation by the Quasi-Biennial Oscillation (QBO) is examined. The composite results show that the suppressed (enhanced) convection anomalies for MJO phases 8-1 (4-5) more concentrated over maritime continent and western Pacific during EQBO (WQBO). As a consequence, more significant wet (dry) anomalies develop in South (eastern) China during MJO phases 8-1 (4-5) configured with easterly (westerly) QBO. The enhancement and expansion of the anomalous tropical convection band do not necessarily correspond to enhancement of the extratropical circulation response to MJO phases 8-1 (4-5) configured with westerly (easterly) QBO. The anomalous high (low) over the maritime continent and western Pacific associated with MJO phases 8-1 (4-5) is intensified (deepened) during easterly (westerly) QBO, leading to large southwesterly (northeasterly) anomalies in South China and coasts, carrying abundant (sparse) moisture. Two anomalous meridional circulation cells are observed for MJO phases 8-1 in the East Asia sector, with downwelling anomalies around 5–20°N, upwelling anomalies around 20–30°N, and another downwelling branch northward of 30°N, which are enhanced during easterly QBO. The anomalous meridional circulation cells are reversed for MJO phases 4-5, which are stronger during westerly QBO with the anomalous downwelling and dry anomalies covering eastern China. The combined impact of MJO phases 8-1 and easterly QBO on the early summer rainfall is noticeable in 1996 and 2020. The enormous rainfall amount appeared along the Yangtze River in 1996 and 2020 due to the extended period of the MJO phases 8-1 under the background of the easterly QBO.

## 1 Introduction

As the dominant mode on the intraseasonal timescale in the tropics, the Madden-Julian Oscillation (MJO) is characterized by eastward-propagating organized convection systems (Madden and Julian, 1971). The MJO is connected with the coupling convection of mixed Rossby-gravity waves (MRGs), which are initialized in the upper level of the western tropical Indian Ocean (Takasuka et al., 2019, 2021). The convection associated with the MJO can excite multiple teleconnections in both the

stratosphere and the troposphere (Garfinkel and Schwartz, 2017; Garfinkel et al., 2014). These teleconnections can further affect the near surface weather and climate (Jenney et al., 2019; Zheng and Chang, 2019). Recent studies also indicate that the stratosphere can also be modulated by the MJO (Garfinkel et al., 2014; Moss et al., 2016; Yang et al., 2019). The stratospheric sudden warming (SSW) and the North Atlantic Oscillation (NAO) can develop following the enhanced convections over the western tropical Pacific (Barnes et al., 2019; Kang and Tziperman, 2018).

The strength of the MJO varies with the season, which is more evident in boreal winter but much weaker in boreal summer (Lafleur et al., 2015; Lu and Hsu, 2017). This difference may be attributed to the surface moistening and strengthened convection in the Intertropical Convergence Zone (ITCZ) in boreal winter, differing from larger static stability and strengthened sinking motion in boreal summer (Wang et al., 2020). This nonuniformity with the season is also identified for the relationship between the Quasi-Biennial Oscillation (QBO) and the MJO, with the MJO-QBO teleconnection getting maximized in boreal winter (Toms et al., 2020; Martin et al., 2021). The teleconnection between QBO and MJO in boreal winters has been widely documented in some recent studies (Densmore et al., 2019; Klotzbach et al., 2019; Kim et al., 2020a; Wang and Wang, 2021). In contrast, the MJO-QBO link in the boreal summer was reported to be weak and have a decadal variability (Yoo and Son, 2016; Wang et al., 2019), although the influence of the MJO on the surface weather in boreal summer has been analyzed ( Zhang et al., 2009; Wang et al., 2013; Bai et al., 2022).

The quasi-biennial oscillation (QBO) is described as periodic alternation of easterly and westerly zonal winds in the tropical stratosphere with an average period of 28 months (Baldwin et al., 2001). As an important phenomenon in stratosphere, QBO can influence surface weather and climate by three routes, including polar stratosphere route (Anstey and Shepherd, 2014; Holton and Tan, 1980, 1982; Rao et al., 2020b), tropical convection route (Collimore et al., 2003; Haynes et al., 2021; Hitchman et al., 2021; Son et al., 2017) and subtropical route (Garfinkel and Hartmann, 2011; Rao et al., 2020a). The anomalous high over the Pacific, owing to the QBO wind downward-arching into the troposphere, influence the Asia-Pacific climate (Rao et al. 2020a; Wang et al., 2021). Hu et al. (2022) found that QBO can also influence summer precipitation in China.

The persistent extreme rainfall (PER) event is a high-impact weather globally, which usually leads to a fast accumulation of water and even urban waterlogging (Wang and Zhang, 2008; Qian et al., 2013; Zou and Ren, 2015; Rao et al., 2022). During June–July PER events occur in East Asia and the rainbelt usually forms from the Yangtze-Huai Rivers to South of Japan, known as the Meiyu-Baiu (Takaya et al., 2020; Takahashi and Fujinami, 2021; Chen et al., 2021a, 2022). On average, the Meiyu-Baiu rain season persists from late June to early July, and the rainfall in June-July has an evident intraseasonal variance with an averaged cycle of 10–20 days (Ding et al., 2020). Previous studies have established the possible relationship between the tropical MJO and the intraseasonal variability of China rainfall in winter (Jia et al., 2011; Ren and Ren, 2017; Chen et al., 2021b), and the tropical MJO can significantly affect the weather in East China (Jeong et al., 2008; Takahashi et al., 2012; Kim et al., 2020b). It is identified that the MJO impact the precipitation in Southern China via exciting a Rossby wave spreading from tropical Indian Ocean to East Asia along the westerly wind waveguide (Zhang et al., 2009).

Considering that the MJO strength can be modulated by the QBO (Densmore et al., 2019; Klotzbach et al., 2019; Wang and Wang, 2021), recent studies have found that the MJO convection is much stronger during the easterly phase of the QBO at 50 hPa than the westerly phase in boreal winter (Son et al., 2017; Toms et al., 2020). When the QBO winds are easterlies at 50 hPa, easterly shears appear below the QBO wind center, which correspond to tropical cold anomalies (and therefore positive meridional temperature gradient anomalies) by the thermal wind balance (Collimore et al., 2003; Rao et al. 2020a).

Therefore, the easterly QBO increases the statistic instability in the upper troposphere, while the westerly QBO decreases the statistic instability. As a consequence, the MJO-related tropical convection enhances in EQBO and weakens in WQBO.

The modulation of QBO on MJO related precipitation in East Asia in boreal winter was reported in Kim et al. (2020a). They found that EQBO enhances the MJO-related rainfall anomalies while WQBO weakens in boreal winter. Given that the rainfall is much larger in boreal summer than in boreal winter (Mao et al., 2022; Wu et al., 2021), a better understanding of

75 the summer rainfall variability is a prerequisite for timely long-range prediction of the weather (Pfahl et al., 2017; Sillmann, 2017). This study is aimed to explore the impact of the tropical MJO on the summer rainfall in China and its modulation by the QBO. An exploration of the impact of the MJO on early summer rainfall and its modulation by the QBO can further improve our understanding of the summer rainfall variability and therefore a better forecast of summer rainfall especially on the long-range timescale (Li, 2016; Zhu et al., 2017; Liang et al., 2019).

This paper is constructed as follows. Following the introduction, section 2 introduces the data and methods employed in this study. Distribution of circulation and rainfall anomalies for typical MJO phases is shown in section 3. The modulation of the QBO on the MJO-related rainfall anomalies during early summer (June-July) in China is discussed in section 4. The physical processes responsible for the eastern China rainfall variability associated with the MJO and its modulation by the QBO are analyzed in section 5. Two typical summers (1996 and 2020) are examined in section 6. Finally, summary and discussion is

presented in section 7.

## 2 Data and methods

To investigate the circulation and rainfall anomalies associated with the tropical MJO, several datasets are used in this study. The European Centre for Medium-Range Weather Forecasts atmosphere reanalysis version 5 (ERA5; Hersbach et al., 2020) is used for construction of the composite circulation and moisture patterns. The surface pressure and outgoing longwave

radiation (OLR), named as top net thermal radiation in ERA5 has a horizontal resolution of 0.25°×0.25° at a single level. Other variables from ERA5 used in this study include the horizontal winds, geopotential height, vertical velocity and specific humidity at pressure levels. Multilevel variables from ERA5 also have a horizonal resolution of 0.25°×0.25° and spans from 1000–1 hPa at 37 pressure levels. In addition, the CPC daily land precipitation (Chen et al., 2008) is employed to calculate the composite rainfall anomalies associated with the MJO. The CPC land precipitation has a horizontal resolution of

0.5°×0.5°, covering the timespan of 1979–2021. To further ensure this relationship, the composite results based on the NCEP/NCAR reanalysis are provided in supplementary Figures S17-S21. Daily interpolated outgoing longwave radiation

(OLR) spanning from 1979–2021 is provided by the National Atmospheric and Oceanic Administration (NOAA) with a horizontal resolution of 2.5°×2.5° (Liebmann and Smith, 1996). Horizontal winds, geopotential height, vertical velocity, surface pressure and specific humidity in NCEP/NCAR Reanalysis I have a horizontal resolution of 2.5°×2.5° (Kalnay et al., 1996). It should be noted that different latitude ranges are chosen for different quantities to better show the key results.

The raw Real-time Multivariate MJO (RMM) index (Wheeler and Hendon, 2004) is used to define the MJO phase. The RMM index is the timeseries of the multivariate empirical orthogonal function (MV-EOF) analysis. The MV-EOF is similar to the traditional EOF analysis but the focused field is not a single variable but a combined one from several different variables (Lee et al., 2013; Li et al., 2019). The variables used for MV-EOF analysis to extract the RMM index include zonal winds at 850 (U850), zonal winds at 200 hPa (U200), and the OLR anomalies meridionally averaged between 15°S and 15°N (Wheeler and Hendon, 2004). The first two principal components (standardized timeseries of the MV-EOFs) are used to define the QBO phases. The two principal components, RMM1 and RMM2 can determine both the MJO strength ($\sqrt[2]{(RMM1)^2 + (RMM2)^2}$) and phase [$\pi \pm$ arctan (RMM2/RMM1)]. The MJO is usually split into eight phases, and every 45° is clustered in one phase (Wheeler and Hendon, 2004). The MJO phases are usually selected only if the amplitude ($\sqrt[2]{(RMM1)^2 + (RMM2)^2}$) is greater than 1.0.

To test the modulation of the QBO on the MJO-related rainfall in Southern China, the QBO index is calculated using the ERA5 zonal winds. The QBO index is defined as the anomalies of zonal mean zonal winds in the deep tropics (10°S-10°N) at 50 hPa. The index in May-June (May-July means show similar results) is used to select the QBO phases: The westerly QBO (WQBO) is defined when the zonal-mean zonal wind anomalies averaged over 10°S-10°N at 50 hPa exceed the 0.7 standard deviations (7 m/s), while the easterly QBO (EQBO) is defined when the zonal-mean zonal wind anomalies fall below -0.7 standard deviations. We also tried to use the threshold of 0.5 standard deviations, and the results are similar but with a lower confidence level than choice of the 0.7 standard deviations. The selected WQBO early summers are 1981, 1983, 1985, 1986, 1988, 1993, 1995, 1999, 2000, 2002, 2009, 2011, 2014, 2017, 2019, and 2021. The selected EQBO early summers are 1982, 1984, 1987, 1992, 1994, 1996, 1998, 2010, 2015, 2016, 2018, and 2020. Note that the early summer is defined as June-July to focus on the influence of the QBO and MJO during the Meiyu period.

### 3 Impact of the tropical MJO on the early summer rainfall in Southern China

Previous studies have indicated that the rainfall anomalies in East Asia is larger and more significant during the MJO phases 8-1 and 4-5 (Zhang et al., 2009). The OLR anomalies and 200 hPa divergence anomalies during MJO phases 8-1 and 4-5 are shown in Figures 1a and 1b. During the MJO phases 8-1, convections are enhanced over the western tropical Indian Ocean but suppressed over the western Pacific and the maritime continent (Figure 1a). As a consequence, anomalous convergence develops over the maritime continent and neighboring areas. In the mirror phases, the convection in the southwest Indian Ocean begins to weaken, while that enhances over most parts of the equatorial Indo-Pacific Oceans, exhibiting a long zonal

band tilting from tropical northwest Indian Ocean to southwest Pacific Ocean (Figure 1b). Anomalous divergence appears along this convection band, maximized around east Indian Ocean and the maritime continent.

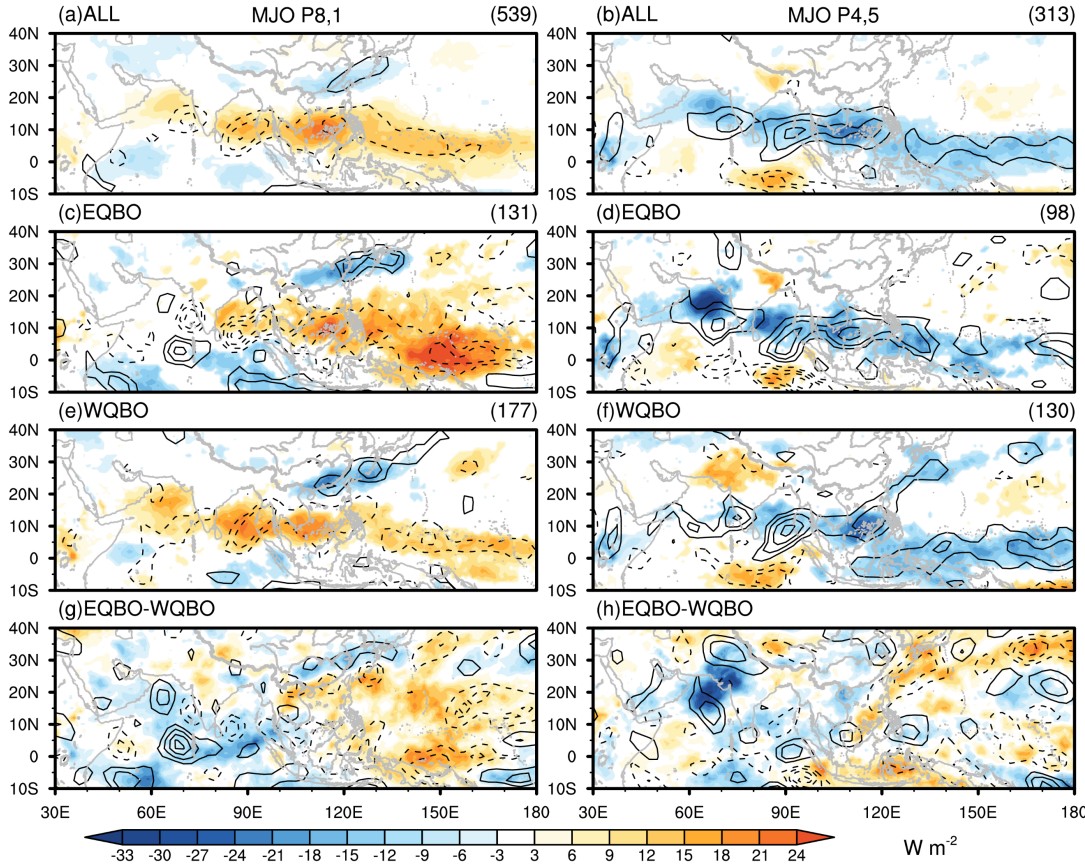

**Figure 1.** Composite OLR (shadings; units: W/m²) and 200-hPa divergence (contours; units: s⁻¹; interval: 3×10⁻⁶) anomalies at the Madden-Julian oscillation (MJO) phases (left) 8-1 and (right) phases 4-5 for (a, b) total days, (c, d) easterly QBO days, (e, f) westerly QBO days and (g, h) EQBO-WQBO difference (dashed lines show convergence anomalies). Only the composite anomalies that are statistically significant at 95% confidence level are shown according to the t-test. The number of days used for each composite map is printed at the top-right corner. The ERA5 reanalysis is shown. The composite OLR and 200-hPa divergence anomalies based on the NCEP/NCAR reanalysis are shown in Figure S17.

To establish the relationship between the MJO and the early summer rainfall variability in China, Figure 2 shows the composite rainfall anomalies during MJO phases 8-1 and 4-5, respectively. Overall, the MJO has a significant impact on rainfall variability in early summer especially over eastern China (Figure 2a, b). Specifically, South China is wetter during the MJO phases 8-1, while central and northeast China is drier with patches of high significance level (Figure 2a). On the contrary, South China is drier during the MJO phases 4-5, while the rainfall anomalies are insignificant in most parts of northern China (Figure 2b). The significant MJO signal in the rainfall variability is not only identified over China but also observed over northern India and Japan, which is beyond the scope of this study.

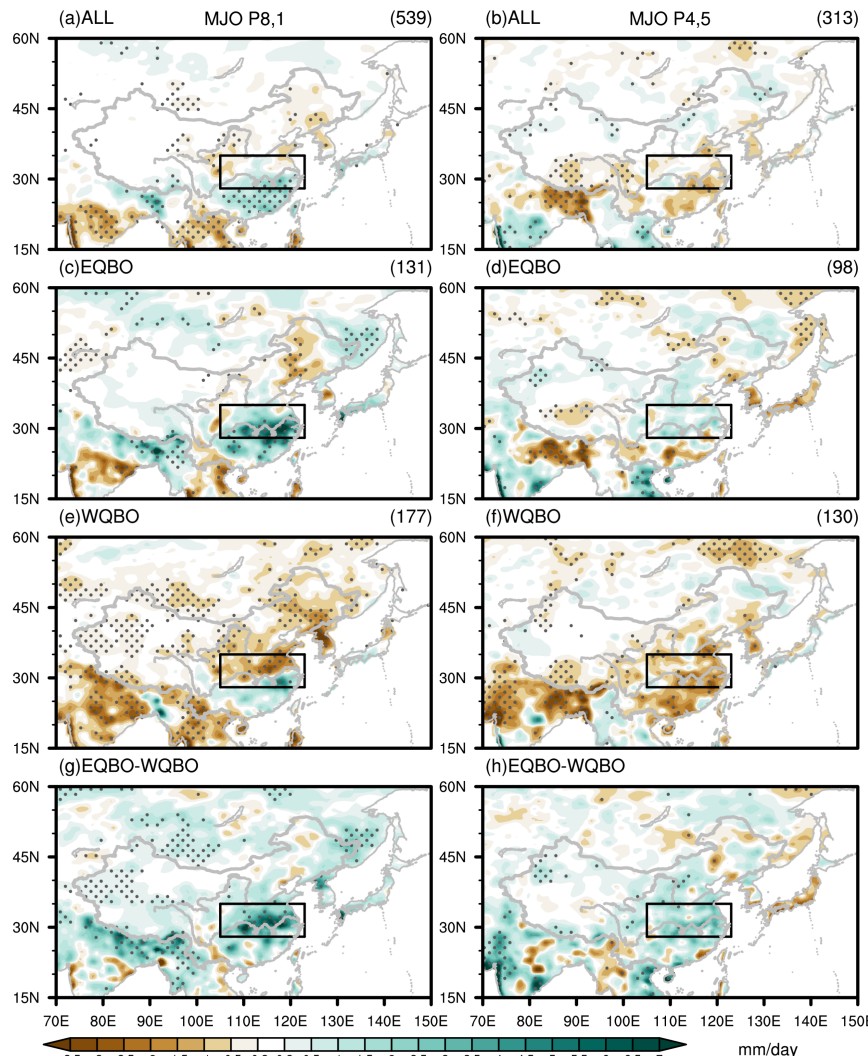

Figure 2. Composite rainfall anomalies (shadings; units: mm/day) at the MJO phases (left) 8-1 and (right) phases 4-5 for (a, b) total days, (c, d) easterly QBO days, (e, f) westerly QBO days and (g, h) EQBO-WQBO difference. The dots denote the composite anomalies at 95% confidence level are shown according to the t-test. The number of days used for each composite map is printed at the top-right corner. The composite rainfall anomalies with the interannual ENSO signals removed are shown in Figure S1.

## 4 Statistical relationship between QBO phases and MJO-related rainfall anomalies

To investigate the modulating role of the QBO for the MJO teleconnection in China rainfall during early summer, the composite for MJO phases 8-1 and 4-5 are also shown separately for EQBO and WQBO (Figures 1c–1f, 2c–2f). Compared with the composite for total days of QBO phases in early summer, the suppressed convection anomalies during MJO phases 8-1 are further localized over the western tropical Indian Ocean and maritime continent during EQBO (Figure 1c). The positive OLR anomalies over western Pacific are wider, and the anomalous convergence at 200 hPa is also more organized.

In contrast, the convection band for the MJO phases 4-5 is uniformly enhanced during EQBO, which is mainly attributed to the increase in the statistic instability in the lower stratosphere and upper troposphere (Rao et al., 2020a). The combined MJO phases 4-5 and EQBO lead to an amplification of the enhanced convection with multiple divergence centers at 200 hPa (Figure 1d).

The tropical convection for the MJO phases 8-1 during WQBO is largely suppressed from the tropical Indian Ocean to the
tropical western Pacific with the positive OLR anomalies much larger over tropical Indian Ocean than the composite for other conditions (Figure 1e). Previous studies have shown that the thermal forcing in the tropical Indian Ocean usually excites a wave train that is reversed from the forced one by the tropical Pacific forcing (Fletcher and Kushner, 2011; Rao and Ren, 2016, 2020). The multiple positive OLR anomaly centers over the western tropical Indian Ocean, Bay of Bengal, maritime continent, and western Pacific might lead to a net weak extratropical circulation response to the MJO, although the
OLR anomalies are amplified. The convections for the MJO phases 4-5 during WQBO are more focused than for other conditions (Figure 1f), and the enhanced convection is mainly located over the Bay of Bengal, maritime continent, and western Pacific. Namely, the enhanced convection over western tropical Indian Ocean is missing for the MJO phases 4-5 configured with the WQBO. Comparing the differences of OLR anomalies between EQBO and WQBO early summers, they have a consistent spatial distribution that convections over the tropical Indian Ocean are enhanced while convections are
suppressed over the tropical Pacific (Figures 1g and 1h), consistent with Gray et al. (2018). This may be due to the QBO-induced cold temperature anomalies extending down to ~100 hPa and a combination of MJO- and QBO-induced reductions in static stability at the tropopause (Abhik and Hendon, 2019; Klotzibach et al., 2019; Rao et al., 2020a).

In short, the configuration of the MJO phases 8-1 (4-5) and EQBO (WQBO) lead to the more localized OLR anomalies over the Maritime continent and neighboring areas. In contrast, the combination of the MJO phases 4-5 (8-1) and EQBO (WQBO)
corresponds to an elongated OLR anomaly band from the western Indian Ocean to the western Pacific with multiple OLR centers. The EQBO enhances the convections over the tropical Indian Ocean and suppresses them over the tropical Pacific in both MJO phases 8-1 and 4-5.

Consistent with the change of the convection associated with the MJO during QBO phases, early summer rainfall anomaly pattern varies (Figure 2c–2f). Specifically, the positive rainfall anomalies in South China for MJO phases 8-1 are enhanced
during EQBO, and negative rainfall anomalies in northeast China (and even northeast Asia) are relatively stable compared with the composite for total days (Figure 2c vs. Figure 2a). In contrast, the composite rainfall anomalies for MJO phases 4-5 are less organized and insignificant during EQBO than the composite for total days (Figure 2d vs. Figure 2b). The positive rainfall anomalies in South China for MJO phases 8-1 are less significant during WQBO, although the dry anomalies in northern and northeast China are enhanced (Figure 2e). In contrast, dry anomalies in South China for MJO phases 4-5 are
strengthened and expand to northern China during WQBO (Figure 2f). The differences of the rainfall anomalies between EQBO and WQBO are consistently enhanced in most of China, and the rainfall anomaly center during MJO phases 8-1 is situated along the Yangzi River (Figures 2g and 2h). The pure composite rainfall anomalies with the ENSO signals removed are shown in Figure S1 to examine the possible interference of ENSO with the composite results. The general patterns for

the OLR composite are nearly unchanged (Figure 1 vs. Figure S1), although the amplitude of anomalous signals is enhanced around the south coast of China after the ENSO signals are removed. The influences of ENSO on the composite EQBO-WQBO difference are also very weak (Figure S1g and S1h).

**5 Physical analysis of the MJO-related rainfall variation and its modulation by the QBO**

The composite horizontal wind and geopotential height anomalies at 850 hPa are shown in Figure 3 for the MJO phases 8-1 and 4-5, respectively. As an anomalous anticyclone (high) develops over positive height anomalies develop from tropical Indian Ocean to northwestern Pacific during the MJO phases 8-1, while negative height anomalies replace during the phases 4-5 (Figure 3a, b). As the tropical convections are suppressed from the Indian Ocean to the western Pacific during the MJO phases 8-1, anomalous easterlies appear over the equator, a band of anomalous high controls the South and Southeast Asia in the lower troposphere. Two high centers are prominent, one over the Bay of Bengal, and the other over the Philippine Islands (Figure 3a). As a consequence, anomalous southwesterlies over South China and the coastal provinces, which carry more abundant moisture. The circulation pattern associated with the MJO phases 8-1 is highly strengthened during EQBO due to the concentrated thermal forcing (Figure 3c). This altered pattern highly resembles the negative phase of the East Asia-Pacific (EAP) pattern with an anomalous high over the Philippines and an anomalous low over northeast Asia. (Nitta, 1987; Li et al., 2018; Xu et al., 2019). In contrast, the circulation pattern for the MJO phases 8-1 is relatively weaker during WQBO (Figure 3e). The anomalous low over northeast Asia is not clearly present, and the anomalous high over the Philippines weakens.

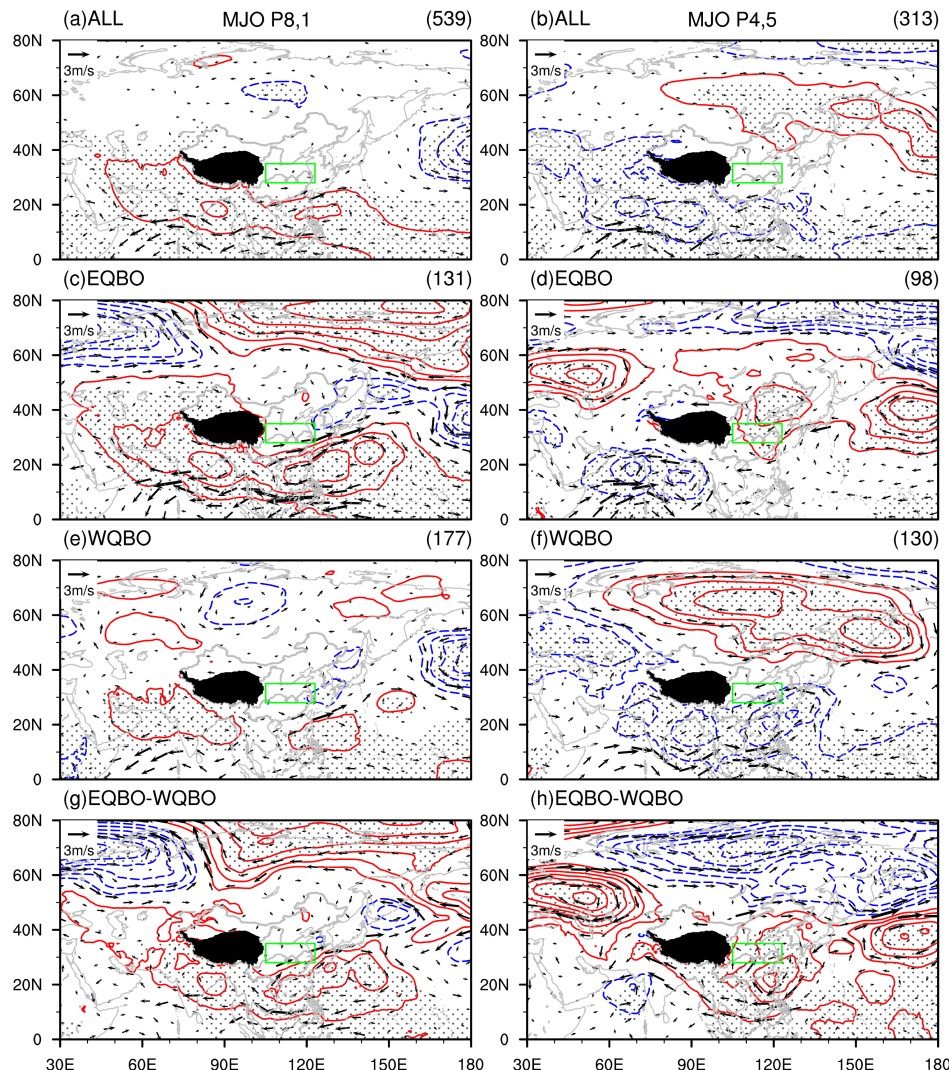

**Figure 3. Composite horizontal wind (vectors; units: m/s) and geopotential height (contours; units: gpm; interval: 5) anomalies at 850 hPa during the MJO phases (left) 8-1 and (right) phases 4-5 for (a, b) total days, (c, d) easterly QBO days, (e, f) westerly QBO days and (g, h) EQBO-WQBO difference. Note that zero contours are omitted, the positive contours are shown in red, and the negative contours are shown in blue. The dots denote the composite height anomalies at 95% confidence level are shown according to the t-test. The number of days used for each composite map is printed at the top-right corner. The ERA5 reanalysis is shown. The composite horizontal wind and geopotential height anomalies based on the NCEP/NCAR reanalysis are shown in Figure S18.**

The circulation pattern at 850 hPa during MJO phases 4-5 is nearly contrary to that during phases 8-1 (Figure 3b vs. 3a). Namely, an anomalous low develops over South and Southeast Asia, while an anomalous high develops from northeast Pacific to North Asia (Figure 3b). This tropical anomalous low associated with the MJO phases 4-5 shrinks in its coverage during EQBO, and significant negative height anomalies are mainly situated from the Arabian Sea to the Bay of Bengal (Figure 3d). Namely, the combination of MJO phases 4-5 and EQBO fail to lead to an enhancement of the circulation variations. In contrast, the tropical circulation anomalies associated with MJO phases 4-5 are stronger during WQBO, when

the convection anomalies are more concentrated. Namely, the anomalous low over the Philippines are stronger with northeasterly anomalies prevailing over coastal South China (Figure 3f). For both MJO phases 8-1 and 4-5, the composite EQBO minus WQBO differences are nearly consistent in low latitudes. An anomalous high appears over Philippines, and anomalous southwesterlies formed over South China during EQBO, although the circulation anomalies at higher latitudes are different (Figures 3g and 3h).

As the two major monsoon systems, the South Asia high (SAH) at 100 hPa and the Western Pacific subtropical high (WPSH) at 500 hPa control the position and intensity of the summer monsoon rainfall band in East Asia (Chen and Zhai, 2016; Guan et al., 2018). The composite of the SAH and WPSH is shown in Figure 4 for the MJO phases 8-1 and 4-5. The SAH is centered over the Tibet in summer (Figure 4a, b), and the WPSH is more sensitive to the MJO phases. The WPSH boundary can extend westward to 117°E during MJO phases 8-1 and 123°E during phases 4-5. Rainfall usually appears to the north of the WPSH, which corresponds to wet anomalies in South China for MJO phases 8-1 and dry anomalies for phases 4-5. The SAH is highly enhanced and extends farther eastward during MJO phases 8-1 configured with the EQBO (Figure 4c). The WPSH is also strengthened and extends farther westward. This configuration creates a favorable condition for South China wetness via enhanced moisture transport in the lower troposphere and intensified divergence in the upper troposphere (shown later). The expansion of the SAH is not evident during MJO phases 8-1 configured with WQBO, and the WPSH also retreats to the ocean and South China Sea.

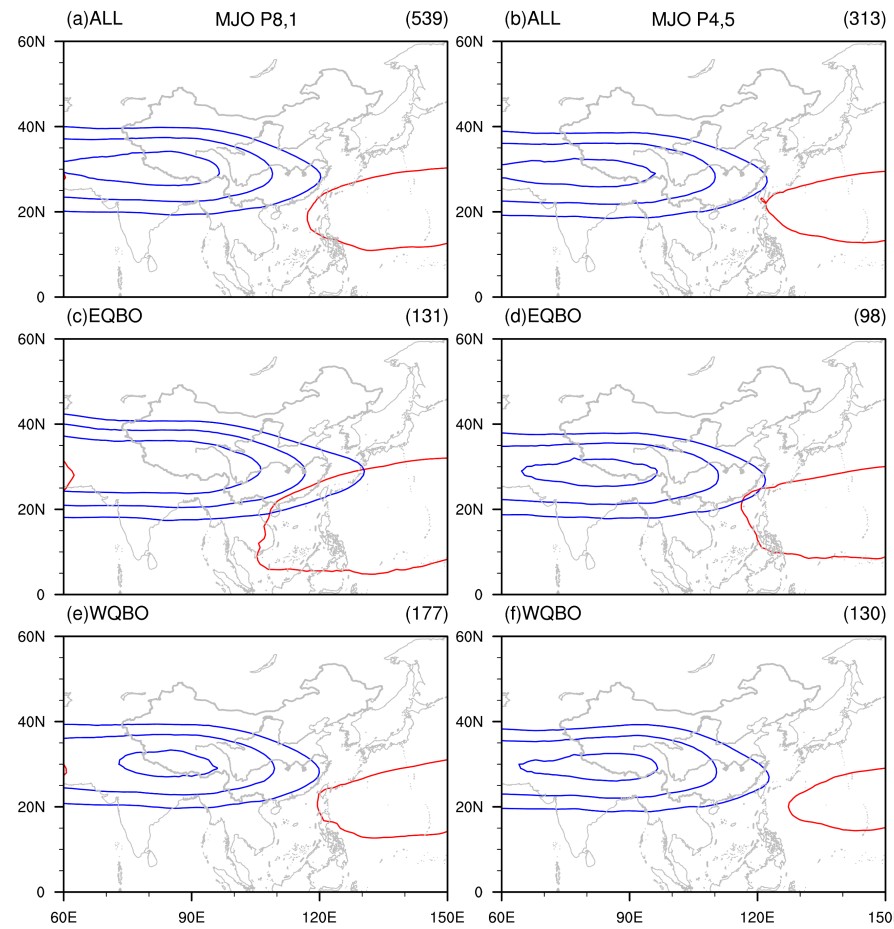

**Figure 4. Composited South Asia High (SAH) at 100 hPa and western Pacific high (WPH) at 500 hPa during the MJO phases (left) 8-1 and (right) phases 4-5 for (a, b) total days, (c, d) easterly QBO days, and (e, f) westerly QBO days. Three height contours (16720, 16760, 16800 gpm) are plotted for the SAH, and one contour (5880 gpm) is plotted for the WPSH. The ERA5 reanalysis is shown. The composite SAH and WPH based on the NCEP/NCAR reanalysis are shown in Figure S19.**

Although the WPSH shrinks in its coverage during the MJO phases 4-5 (Figure 4b), it can still be modulated by the phase of the QBO. It is observed that the WPSH extends westward to South China Sea and covers the coast of South China during MJO phases 4-5 configured with EQBO, which weakens the draught of South China (Figure 4d). The SAH is changed little during MJO phases 4-5 from EQBO to WQBO (Figure 4f). In contrast, the WPSH retreats to the ocean, which corresponds to the enhancement of the draught in South China.

To better understand the rainfall anomalies during the MJO phases 8-1 and 4-5, the vertical cross-sections of the meridional circulation averaged from 110–120°E are shown in Figure 5. As the convection is suppressed over the South China Sea during the MJO phases 8-1, anomalous downwelling develops over the northern tropics from 0–20°N. The convergence is clearly present at 200 hPa and 10°N, and anomalous upwelling appears over 20–30°N, which explains the wetness of South China (Figure 5a). The anomalous downwelling is also present from 30–40°N, which corresponds to the draught of northern

China. Namely, two meridional secondary circulation cells are clearly present, one from 10–25°N, and the other from 25–35°N. With the WPSH and SAH approaching closer during MJO phases 8-1 configured with the EQBO, the anomalous convergence in lower troposphere and the anomalous divergence in upper troposphere are strengthened, and the anomalous northerlies are enhanced in the lower troposphere (Figure 5c). In contrast, the anomalous upwelling in the lower troposphere evidently weakens, and the upwelling band narrows during MJO phases 8-1 configured with WQBO (Figure 5e).

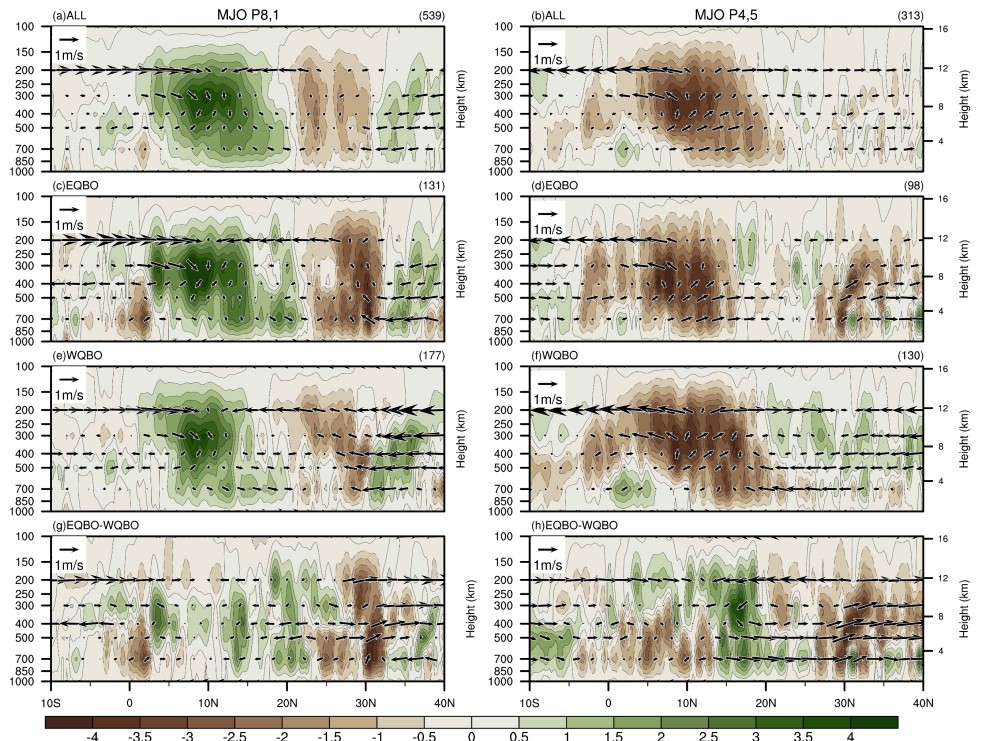

**Figure 5. Composited latitude-pressure cross section of meridional circulation anomalies (vectors; units: m/s, Pa/s) and vertical velocity anomalies (shadings; units: 0.01 Pa/s) averaged from 110–120°E during the MJO phases (left) 8-1 and (right) phases 4-5 for (a, b) total days, (c, d) easterly QBO days, (e, f) westerly QBO days and (g, h) EQBO-WQBO difference (brown denotes upward motion). The vertical velocity anomalies have been multiplied by -10 to better show the vectors. The ERA5 reanalysis is shown. The composite meridional circulation anomalies based on the NCEP/NCAR reanalysis are shown in Figure S20.**

The anomalous tropical vertical motion during MJO phases 4-5 (Figure 5b) is nearly opposite to that during phases 8-1. As the convection in the tropics is enhanced, anomalous upwelling occurs in wide regions. The anomalous downwelling is only seen in the lower troposphere around 25°N and 30°N. The anomalous downwelling is even narrower and weaker over South China when MJO phases 4-5 are configured with EQBO (Figure 5d), which corresponds to insignificant rainfall anomalies in

South China (Figure 2d). In contrast, the anomalous downwelling in eastern China is enhanced during MJO phases 4-5 configured with WQBO (Figure 5f), explaining the dry anomalies (Figure 2f). For both MJO phases 8-1 and 4-5, the EQBO minus WQBO difference shows consistent anomalous downward motion around 15-20°N and upward motion around 30°N (Figures 5g and 5h), consistent with the anomalous circulation in lower troposphere (Figures 3g and 3h).

In order to examine the moisture conditions associated with the MJO, the vertically integrated moisture flux (VIMF) anomalies from 1000–300 hPa and the VIMF convergence are shown in Figure 6. The VIMF is the thickness-weighted sum of the product for humidity and horizontal wind. Significant anomalous VIMF convergence anomalies appear in coastal South China during MJO phases 8-1, and weak anomalous VIMF divergence appears in parts of northern China (Figure 6a). This pattern intensifies during EQBO: significant anomalous VIMF convergence appears over South China, contrasted with anomalous VIMF divergence in tropics (Figure 6c). As the strong VIMF band moves farther eastward during WQBO, the anomalous VIMF convergence also moves eastward (Figure 6e), reminiscent of the horizontal wind anomalies at 850 hPa (Figure 3e).

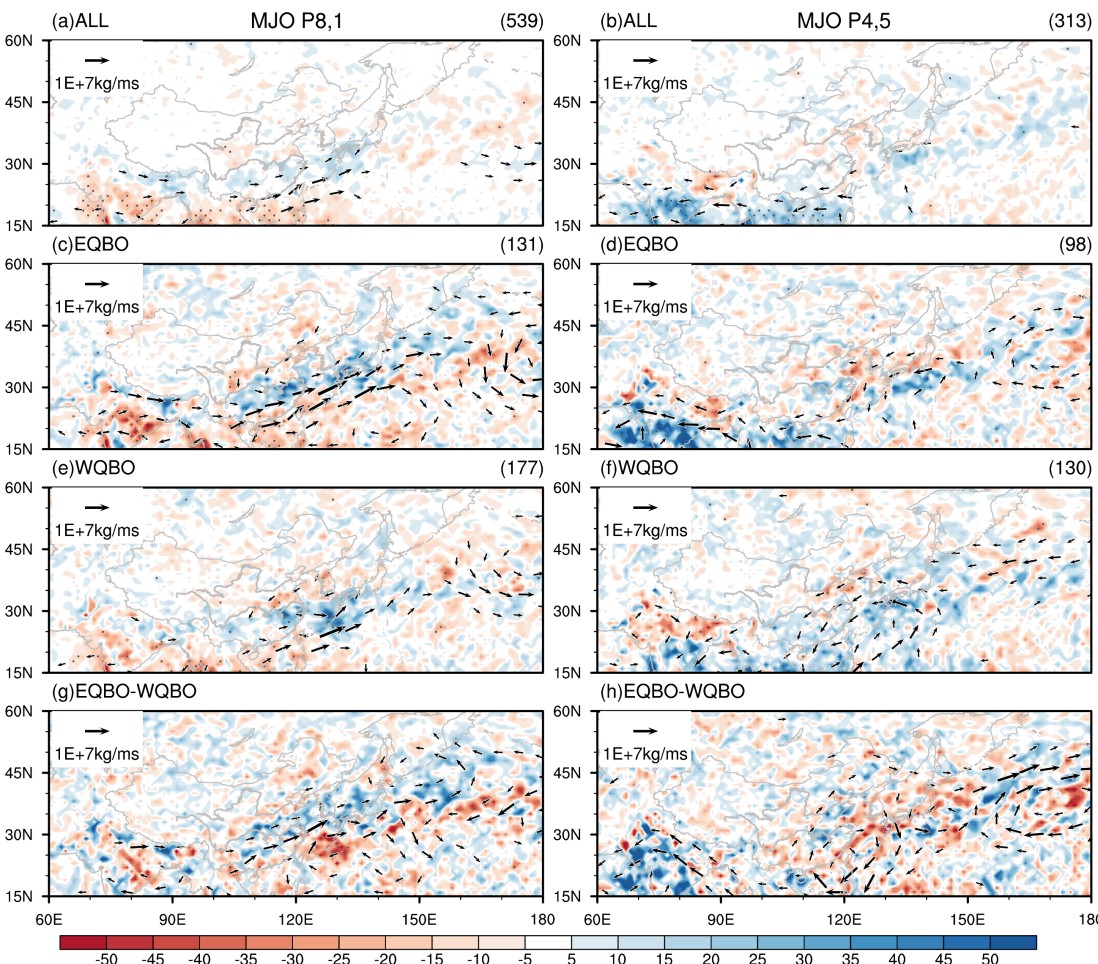

**Figure 6. Composited vertically integrated moisture flux (VIMF) anomalies (vectors; units: $10^2$ kg m$^{-1}$ s$^{-1}$) and its horizonal convergence (VIMFC) anomalies (shadings; units: kg m$^{-2}$ s$^{-1}$) during the MJO phases (left) 8-1 and (right) phases 4-5 for (a, b) total days, (c, d) easterly QBO days, (e, f) westerly QBO days and (g, h) EQBO-WQBO difference. Note that only the VIMF lager than $0.5 \times 10^2$ kg m$^{-1}$s$^{-1}$ are shown. The VIMFC anomalies that are statistically significant at the 95% confidence level are dotted. The ERA5 reanalysis is shown. The composite VIMF and VIMFC anomalies based on the NCEP/NCAR reanalysis are shown in Figure S21.**

The anomalous moisture divergence is observed over parts of East China during MJO phases 4-5, as the anomalous moisture convergence is observed over South China Sea (Figure 6b). The VIMF and VIMFC patterns associated with the MJO phases 4-5 change limitedly during the EQBO (Figure 6d), while the patterns are enhanced during WQBO (Figure 6e), explaining the significant draught anomalies in eastern China for the MJO phases 4-5 together with the WQBO. Although there are some differences between MJO phases 8-1 and 4-5, it can be observed that anomalous VIMF divergence around the south coast of China and anomalous VIMF convergence around north of the Yangtze River are enhanced in EQBO as compared with WQBO (Figures 6g and 6h). In addition, VIMF is also enhanced over South China due to the strengthening of the local southwesterlies. This moisture transport pattern is consistent with the anomalous meridional circulation pattern for the composite EQBO-WQBO difference (Figures 5g and 5h).

## 6 Three-case studies

The Meiyu-Baiu is a rainy period in East Asia and has a significant impact on the agricultural growth and production, human societies, ecosystems, and the natural environment (Rao et al., 2022; Feng et al., 2007). Extreme precipitation events take place during Meiyu-Baiu period, which can increase the river flow and lead to heavy economic losses and even high death tolls (Rao et al., 2022; Qian et al., 2013). The MJO can modulate the East Asian summer rainfall, and extreme rainfall events in eastern China usually appear during certain MJO phases (Liang et al., 2021; Wang et al., 2022). Three cases are selected to further verify the possible impact of the MJO phases 8-1 on eastern China heavy rainfall during EQBO. The total days of the MJO phases 8-1 exceed 20% of all days in June-July for those three years (1996: 21%; 2016: 36%; 2020: 54%). The Meiyu-Baiu average cumulative rainfall exceeded 600 mm in 1996, 2016 and 2020, causing enormous economic losses in China.

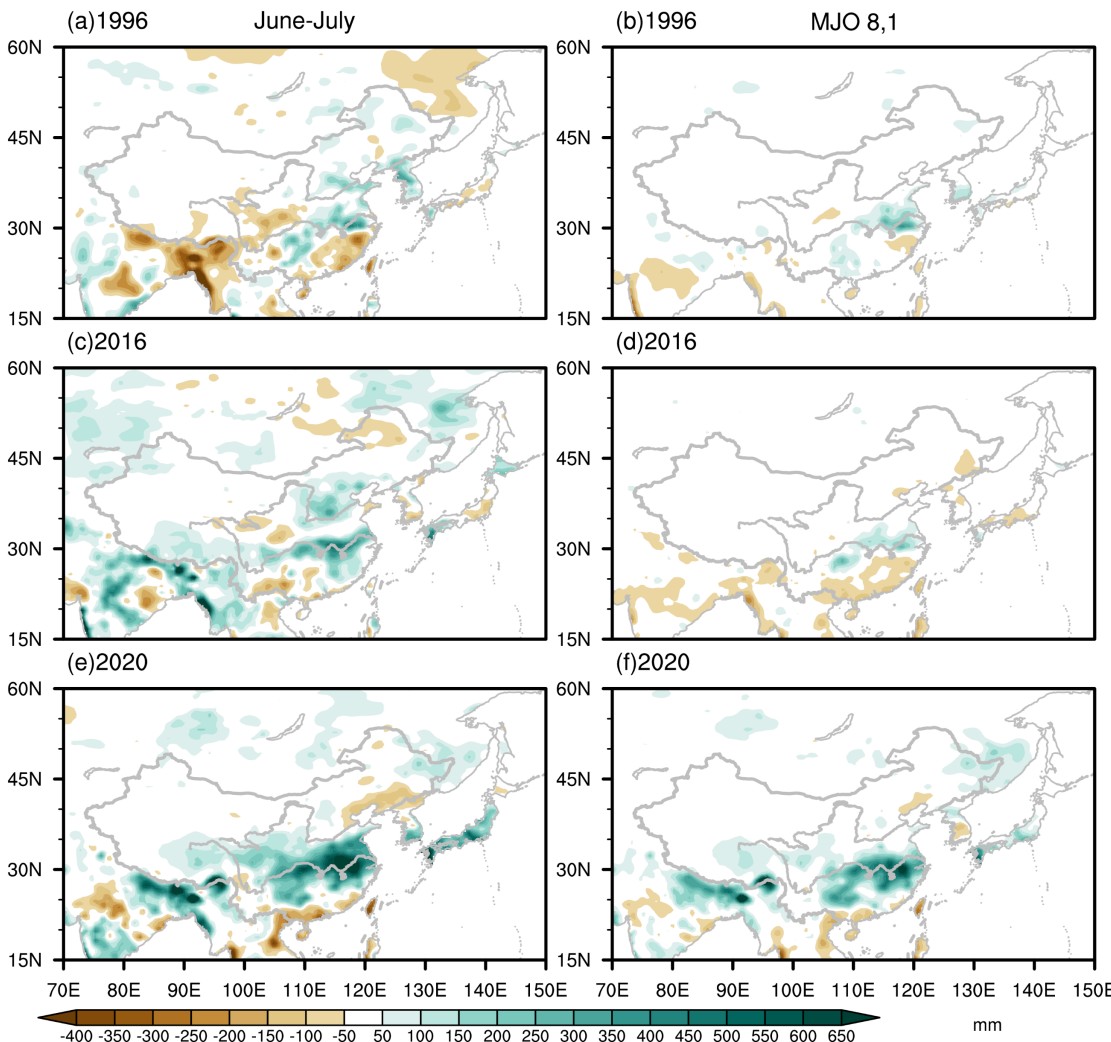

**Figure 7. Total precipitation anomalies (units: mm) in June-July (left) and contributions by MJO phases 8-1 (right) for (a, b) 1996, (c, d) 2016, and (e, f) 2020.**

The composite rainfall anomalies during June-July for three typical case years are shown in Figure 7. Eastern China is characterized by more rainfall in the south and less rainfall in the north (Figure 2a). The wet anomalies mainly develop along the Yangtze River and its south flanks. Compared to the composite of MJO phases 8-1 for all years, the Meiyu-Baiu rainfall band in 1996, 2016 and 2020 is biased further northward (Figure 7a, c, e). Recent studies have identified that the MJO phases 1 and 2 have a significant impact on the 2020 extreme Meiyu rainfall (Liang et al., 2021; Zhang et al., 2021). The composite for MJO phases 1-2 is shown in Figure S16, and similar QBO modulation is also observed. With the phase of the QBO considered, the rainfall amplitude associated with the MJO can be further intensified in the composite and further verified by the three case years in 1996, 2016 and 2020 (Figure 7b, d, f).

## 7 Summary and discussions

The possible impact of the MJO on East Asian winter rainfall and its modulation by the QBO have been widely reported in literature, and the modulation of the East Asian summer rainfall–MJO relation is still not well understood. This study evaluates the relationship between the MJO teleconnection and eastern China rainfall in early summers and its sensitivity to the QBO phase. The main findings in the study are as follows.

I.    The composite convection (OLR) anomaly amplitude can be modulated by the QBO phase. Specifically, the positive
OLR anomaly magnitude associated with the MJO phases 8-1 is strengthened especially over tropical Indian Ocean during WQBO, and the suppression convection band covers tropical Indian Ocean, maritime continent, and western Pacific. The positive OLR anomalies for MJO phases 8-1 weaken especially over tropical Indian Ocean and are more concentrated and mainly develop over maritime continent and western Pacific during EQBO. Similarly, the negative OLR anomaly (enhanced convection) band for MJO phases 4-5 extends from tropical Indian Ocean to western Pacific.
The negative OLR anomalies for MJO phases 4-5 are amplified especially over tropical Indian Ocean during EQBO, and they weaken and become more concentrated over maritime continent and western Pacific.

II.    Eastern China rainfall in early summer is significantly influenced by the MJO. The composite MJO-related rainfall pattern shows that South China is wetter during MJO phases 8-1 with a high significance level, while parts of northern China are drier. Although the tropical convection variations are larger for the MJO phases 8-1 configured with WQBO
than that configured with EQBO, the wetness over South China and the Yangtze River Valley is more evident for EQBO than for WQBO. Similarly, although the negative OLR band is wider for the MJO phases 4-5 configured with EQBO than that configured with WQBO, the drought anomalies over eastern China are broader for WQBO than for EQBO.

III.    Two Asian monsoon systems (SAH and WPSH) show somewhat sensitivity to the MJO and QBO phases. The SAH is
wide and expands eastward for MJO phases 8-1 configured with EQBO, and meanwhile the WPSH expands further westward to South China Sea. On the contrary, the SAH and/or WPSH size is smaller and the intensity is weaker for MJO phases 4-5 configured with WQBO than other conditions. With the change of the SAH and WPSH, the moisture flux divergence or convergence anomalies are more evident for the two configurations.

IV.    The negative phase of East Asia-Pacific (EAP) pattern or the so-called Pacific-Japan (PJ) pattern is observed in MJO
phases 8-1 configured with EQBO, while the positive EAP/PJ pattern is clearly present at 850 hPa in MJO phases 4-5 configured with WQBO. Two anomalous meridional circulation cells are observed for MJO phases 8-1 in the East Asia sector, with significant 200-hPa convergence - tropospheric downwelling anomalies around 5–20°N, 200-hPa divergence - upwelling anomalies around 20–30°N, and another downwelling branch northward of 30°N. These two anomalous meridional circulation cells for MJO phases 8-1 are enhanced during EQBO, corresponding to the more
significant wet anomalies in South China. The anomalous meridional circulation cells are reversed for MJO phases 4-5, which are stronger during WQBO, with the anomalous downwelling and dry anomalies covering eastern China.

V.   The combined impact of MJO phases 8-1 and EQBO on the early summer rainfall is noticeable for some typical cases. The enormous rainfall amount appeared along the Yangtze River in 1996, 2016 and 2020 due to the extended period of the MJO phases 8-1 configured with EQBO.

The enhancement and expansion of the tropical maximum convection does not necessarily correspond to strengthened extratropical circulation response. Consistent with MJO-related variations of early summer precipitation in eastern China, the anomalous high (low) over the maritime continent and western Pacific associated with MJO phases 8-1 (4-5) is heightened (deepened) during EQBO (WQBO) when compared with WQBO (EQBO). As a consequence, large southwesterly anomalies prevail in South China and coasts when MJO phases 8-1 are configured with EQBO, carrying abundant moisture.

Northeasterly anomalies prevail in the lower troposphere over eastern China and drought occur in eastern China when MJO phases 4-5 are configured with WQBO.

The QBO impact the summer rainfall mainly via the tropical convection pathways, as the stratospheric polar pathway is more evident in the Atlantic-Europe sector in winter (Rao et al., 2020a, 2023). The tropical static instability usually enhances in the lower stratosphere and upper troposphere especially over the Indo-Pacific Oceans during EQBO (e.g. Gray et al., 2018;

Klotzbach et al., 2019), which modulate the strength and area of the MJO-related convection over western Pacific and South China Sea (Densmore et al., 2019; Klotzbach et al., 2019). Further, the origination of MJO can also be modulated by the QBO (Toms et al., 2020). Besides, previous studies also found that the MJO can also affect the extratropical stratosphere especially in winter (Alexander et al., 2018; Garfinkel and Schwartz, 2017). Due to a pause of the extratropical stratospheric pathway in summer, the MJO-stratosphere links are worth of exploring for a better understanding of the East Asian climate

variability in winter. However, the larger area of the convection anomalies does not necessarily correspond to larger circulation anomalies in the Asia-Pacific sector and rainfall anomalies in eastern China. Our results find that the concentrated convection anomalies in the tropics probably have a larger impact on the East Asian climate.

The combined impact of the QBO and MJO on East Asian rainfall in early summer is different from that in winter. Kim et al. (2020a) reported that EQBO amplifies the anomalous rainfall associated with the MJO in winter. This study finds that

EQBO is favorable for increase of rainfall associated with MJO phases 8-1, while WQBO is favorable for decrease of rainfall associated with MJO phases 4-5. The seasonal differences in the QBO modulation for the MJO-related rainfall variation is likely related to the seasonal changes in the tropical mean state.

Liang et al. (2021) found that the eastward motion of the MJO is difficult during La Nina than El Nino. Moreover, both the El Nino-Southern Oscillation (ENSO) and QBO phases can impact the eastward motion of the MJO (Sun et al., 2019; Huang

and Pegion, 2022). Huang and Pegion (2022) pointed out that La Nina-like cold sea surface temperature (SST) anomalies can weaken the westward-propagating wave activity and confine it in western Pacific, leading to more standing MJO events. The impact of ENSO on the MJO events has been considered in some recent reports (Sun et al., 2019). This study also considers the possible interference of ENSO in the composite for MJO, but a preclusion of the interannual signals associated with ENSO does not significantly change the composite for MJO configured with the QBO (Figure S1).

The tropical intraseasonal oscillation (ISO) has different propagation patterns between boreal summer and winter. The Boreal Summer ISO (BSISO) mode with prominent northward propagation and large variability in off-equatorial monsoon trough regions is the prominent mode in boreal summer (Kikuchi et al., 2012). Although the amplitude of the MJO weakens during the boreal summer season, QBO is identified to affect the MJO-related convection and circulation in boreal summer based on statistical analysis in this research. Besides, the modulation of QBO on the BSISO-related convection and

circulation is also inspected. The QBO has a consistent influence on the BSISO-related convection and circulation with MJO in boreal summer (Figures S2-S13; BSISO1: Figures S2-S7; BSISO2: Figures S8-S13) using the phase division by Lee et al. (2013). The modulation of QBO on BSISO-related convection and precipitation is similar to that for MJO phases. However, the SAH is not clearly modulated by the QBO phases for both BSISO1 and BSISO2. The WPH in BSISO1 phases 8 and 1 does not appear in WNP, which is inconsistent with the typical Meiyu circulation pattern.

The QBO-MJO link in boreal winter is reported to intensify in recent years and near future (Kim et al., 2020a). Recent studies also reported that the QBO teleconnections in winter are likely to enhance in the future (Rao et al., 2020b; 2023). This study provides evidence that the MJO teleconnection is also clearly present in early summer, but the future change of the MJO teleconnection in early summer is still unexplored. An evaluation of the state-of-the-art models in reproducing the MJO-China rainfall linkage and its sensitivity to the QBO is left for future study. With high-skill models selecting from the

Coupled Model Intercomparison Project, a more confident projection of the MJO teleconnection is also possible in the follow-up study.

**Acknowledgments**

This research was funded by the National Natural Science Foundation of China (grant nos. 42030605 and 42175069). The CPC land daily precipitation data are available from their website (https://psl.noaa.gov/data/gridded/data.cpc.globalprecip.

html). The OLR data are provided by the NOAA (https://psl.noaa.gov/data/gridded/data. olrcdr.interp.html). The ERA5 reanalysis data are provided by the ECMWF (https://cds.climate.copernicus.eu/). The Real-time Multivariate MJO (RMM) index data are provided by the Bureau of Meteorology Australia (BoM) (http://www.bom.gov.au/climate/mjo/). The NCEP/NCAR Reanalysis 1 data are derived from the NOAA (https://psl.noaa.gov/data/gridded/data.ncep.reanalysis.html).

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
