# Peer review of "Modulation of the Intraseasonal Variability of Early Summer Precipitation in Eastern China by the Quasi-Biennial Oscillation and the Madden-Julian Oscillation"

_EGUsphere, 2023_

## Author Comment (AC1)

Atmospheric Chemistry and Physics

Supporting Information for

[revised manuscript text omitted]

**Figure S17. Composite horizontal wind (vectors; units: m/s) and geopotential height (contours; units: gpm; interval: 5) anomalies at 850 hPa using the NCEP-NCAR reanalysis during the MJO phases (left) 8-1 and (right) phases 4-5 for (a, b) total days, (c, d) easterly QBO days, (e, f) westerly QBO days and (g, h) EQBO-WQBO difference. Note that zero contours are omitted, the positive contours are shown in red, and the negative contours are shown in blue. The dots denote the composite height anomalies at 95% confidence level according to the t-test. The number of days used for each composite map is printed at the top-right corner.**

[Figure]

**Figure S18. Composited South Asia High (SAH) at 100 hPa and western Pacific high (WPH) at 500 hPa using the NCEP-NCAR reanalysis during the MJO phases (left) 8-1 and (right) phases 4-5 for (a, b) total days, (c, d) easterly QBO days, and (e, f) westerly QBO days. Three height contours (16720, 16760, 16800 gpm) are plotted for the SAH, and one contour (5880 gpm) is plotted for the WPSH.**

[Figure]

**Figure S19.** Composited vertically integrated moisture flux (VIMF) anomalies (vectors; units: $10^7$ kg m$^{-1}$ s$^{-1}$) and the VIMF horizontal convergence (VIMFC) anomalies (shadings; units: kg m$^{-2}$ s$^{-1}$) using the ERA5 reanalysis during the MJO phases (left) 8-1 and (right) phases 4-5 for (a, b) total days, (c, d) easterly QBO days, (e, f) westerly QBO days and (g, h) EQBO-WQBO difference. Note that only the VIMF anomalies lager than $0.5\times10^7$ kg m$^{-1}$s$^{-1}$ are shown. The VIMFC anomalies that are statistically significant at the 95% confidence level are dotted.

---

## Author Response (AR1)

**Response to Reviewer # 1**

By analyzing reanalysis and observational data, this study suggests phases 8-1 and phases 4-5 of the Madden-Julian Oscillation (MJO) have different influences on early-summer precipitation over South China, with a particular focus on the modulation of the Quasi-Biennial Oscillation (QBO). It is an intriguing topic and may have substantial implications for S2S forecasting.
Response: Thank you for your positive comments.

Although some valuable phenomena have been discovered, this manuscript lacks a clear explanation of how QBO affects tropical convection anomalies and extratropical atmospheric circulations. As the influence of the Boreal Summer Intraseasonal Oscillation (BSISO) or the summer MJO on East Asian/East China rainfall has been extensively studied in previous studies, the authors should make more effort to explain the role of QBO in modulating the MJO-related tropical convection and the underlying mechanisms.
Response: We agree that we should first explore the role of QBO in modulating the MJO-related tropical convection. The QBO mainly impact the tropical convection by changing the statistic instability in the lower stratosphere and upper troposphere. When the QBO winds are easterlies at 50 hPa, easterly shears appear below the QBO wind center, which correspond to tropical cold anomalies (and therefore positive meridional temperature gradient anomalies) by the thermal wind balance (Collimore et al., 2003; Rao et al. 2020a). Therefore, the easterly QBO increases the statistic instability in the upper troposphere, while the westerly QBO decreases the statistic instability. As a consequence, the MJO-related tropical convection enhances in EQBO and weakens in WQBO.
We added the EQBO-WQBO difference in Figures 1, 2, 3, 5 and 6 to show how QBO affects tropical convection anomalies and extratropical atmospheric circulations. We added a discussion about the mechanism in Introduction and section 4:

- "When the QBO winds are easterlies at 50 hPa, easterly shears appear below the QBO wind center, which correspond to tropical cold anomalies (and therefore positive meridional temperature gradient anomalies) by the thermal wind balance (Collimore et al., 2003; Rao et al. 2020a). Therefore, the easterly QBO increases the statistic instability in the upper troposphere, while the westerly QBO decreases the statistic instability. As a consequence, the MJO-related tropical convection enhances in EQBO and weakens in WQBO." (Page 3 Lines 71-75)

- "This may be due to the QBO-induced cold temperature anomalies extending down to ~100 hPa and a combination of MJO- and QBO-induced reductions in static stability at the tropopause (Abhik and Hendon, 2019; Klotzibach et al., 2019; Rao et al., 2020a)." (Page 7 Lines 172-174)

Despite widespread recognition of the MJO-QBO link in recent studies, it is generally accepted that the link is robust and strong only during boreal winter (e.g., Martin et al. 2021). For example, Yoo and Son (2016) found no linear correlation between QBO and MJO activity in summer; Wang et al. (2019) suggested that the QBO-MJO link in

summer is weak and has decadal variability. The seasonal dependence of the MJO-QBO link is hypothesized to be caused by differences in its amplitude, location, and propagation during different seasons.

Response: The MJO-QBO link is actually weak in the entire boreal summer; however, this relationship is statistically evident in boreal early summer (June-July). We identified that the MJO-QBO link plays an important role in rainfall anomalies during Meiyu-Baiyu period. To supplement the background as you are concerned, we added a discussion in Introduction:

- "This nonuniformity with the season is also identified for the relationship between the Quasi-Biennial Oscillation (QBO) and the MJO, with the MJO-QBO teleconnection getting maximized in boreal winter (Toms et al., 2020; Martin et al., 2021). The teleconnection between QBO and MJO in boreal winters has been widely documented in some recent studies (Densmore et al., 2019; Klotzbach et al., 2019; Kim et al., 2020a; Wang and Wang, 2021). In contrast, the MJO-QBO link in the boreal summer was reported to be weak and have a decadal variability (Yoo and Son, 2016; Wang et al., 2019), although the influence of the MJO on the surface weather in boreal summer has been analyzed (Zhang et al., 2009; Wang et al., 2013; Bai et al., 2022)." (Page 2 Lines 41-49)

First, the authors should examine the significance of the differences in tropical convection and extratropical atmospheric circulations between the EQBO and WQBO years, instead of the significance of their respective differences from climatology as shown in the current manuscript.

Response: We added the EQBO-WQBO difference in Figures 1, 2, 3, 5 and 6 to show how QBO affects tropical convection anomalies and extratropical atmospheric circulations. We added the description of the EQBO-WQBO difference for some figures:

- "Comparing the differences of OLR anomalies between EQBO and WQBO early summers, they have a consistent spatial distribution that convections over the tropical Indian Ocean are enhanced while convections are suppressed over the tropical Pacific (Figures 1g and 1h), consistent with Gray et al. (2018)." (Page 7 Lines 170-172)

- "The differences of the rainfall anomalies between EQBO and WQBO are consistently enhanced in most of China, and the rainfall anomaly center during MJO phases 8-1 is situated along the Yangzi River (Figures 2g and 2h)." (Page 7 Lines 187-189)

- "For both MJO phases 8-1 and 4-5, the composite EQBO minus WQBO differences are nearly consistent in low latitudes. An anomalous high appears over Philippines, and anomalous southwesterlies formed over South China during EQBO, although the circulation anomalies at higher latitudes are different (Figures 3g and 3h)." (Page 10 Lines 222-225)

- "For both MJO phases 8-1 and 4-5, the EQBO minus WQBO difference shows consistent anomalous downward motion around 15-20°N and upward motion

around 30°N (Figures 5g and 5h), consistent with the anomalous circulation in lower troposphere (Figures 3g and 3h)." (Page 12 Lines 268-270)

- "Although there are some differences between MJO phases 8-1 and 4-5, it can be observed that anomalous VIMF divergence around the south coast of China and anomalous VIMF convergence around north of the Yangtze River are enhanced in EQBO as compared with WQBO (Figures 6g and 6h). In addition, VIMF is also enhanced over South China due to the strengthening of the local southwesterlies. This moisture transport pattern is consistent with the anomalous meridional circulation pattern for the composite EQBO-WQBO difference (Figures 5g and 5h)." (Page 14 Lines 289-294)

Next, it is necessary to show in detail how QBO impacts summertime MJO-related tropical convection anomalies. (Some mechanisms have been proposed for the MJO-QBO link in winter, including temperature stratification, wind shear, and cloud-radiative feedback).

Response: Abhik and Hendon (2019) proposed that a combination of MJO- and QBO-induced reductions in static stability at the tropopause plays an important role in strengthening MJO convection during EQBO. The QBO easterlies in the lower stratosphere overlay colder temperatures down to the tropopause in order to maintain thermal wind balance (Son et al., 2017). Hence, EQBO is associated with a destabilized equatorial tropopause region. Klotzbach et al. (2019) proposed the MJO is strengthened during EQBO because the QBO-induced cold temperature anomaly extends down to ~100 hPa and constructively adds to the destabilization. We added the explanation in the introduction:

- "This may be due to the QBO-induced cold temperature anomalies extending down to ~100 hPa and a combination of MJO- and QBO-induced reductions in static stability at the tropopause (Abhik and Hendon, 2019; Klotzibach et al., 2019; Rao et al., 2020a)." (Page 7 Lines 172-174)

My second concern is ENSO's effect. The El Niño-Southern Oscillation (ENSO) has a nonlinear effect on East Asian summer monsoon precipitation, and ENSO can interact with QBO in many ways. Some studies even speculate that the summer MJO-QBO link in observations but not in simulations is a consequence of ENSO modulation (e.g., Wang et al. 2019). How did the authors remove ENSO effects in this study (L332-334)? A detailed explanation of this aspect is necessary.

Response: We used the ENSO index to regress the interannual ENSO signals, and the ENSO-related signals are then removed from the data. The composite OLR for MJO with the interannual ENSO signals removed is shown in Figure S1. To address your concern, we also added a discussion:

"The pure composite rainfall anomalies with the ENSO signals removed are shown in Figure S1 to examine the possible interference of ENSO with the composite results. The general patterns for the OLR composite are nearly unchanged (Figure 1 vs. Figure S1), although the amplitude of anomalous signals is enhanced around the south coast of China after the ENSO signals are removed. The influences of ENSO on the

composite EQBO-WQBO difference are also very weak (Figure S1g and S1h)." (Page 8 Lines 189-193)

In addition to meridional circulation changes (Fig. 5), Meiyu rainfall is also associated with northward-propagating atmospheric teleconnections triggered by tropical convection anomalies. Perhaps this issue needs to be examined and discussed in more detail.

Response: Northward-propagating atmospheric teleconnections triggered by tropical convection anomalies can be observed in meridional circulation anomalies (Figure 5) and geopotential height anomalies (Figure 3). The negative phase of East Asia-Pacific (EAP) pattern or the so-called Pacific-Japan (PJ) pattern is observed in MJO phases 8-1 configured with EQBO, while the positive EAP/PJ pattern is clearly present at 850 hPa in MJO phases 4-5 configured with WQBO. To well address your concern, we added some discussion in the summary section:

- "The negative phase of East Asia-Pacific (EAP) pattern or the so-called Pacific-Japan (PJ) pattern is observed in MJO phases 8-1 configured with EQBO, while the 850hPa circulation anomalies in MJO phases 4-5 configured with WQBO are similar to the positive EAP pattern, while the positive EAP/PJ pattern is clearly present at 850 hPa in MJO phases 4-5 configured with WQBO." (Page 17 Lines 357-359)

The case study of 1996 and 2020 is kind of rough and pointless. I have some relevant suggestions and questions.

Response: We revised according to your suggestions.

- "Three cases are selected to further verify the possible impact of the MJO phases 8-1 on eastern China heavy rainfall during EQBO. The total days of the MJO phases 8-1 exceed 20% of all days in June-July for those three years (1996: 21%; 2016: 36%; 2020: 54%)." (Page 14 Lines 300-302)

  1. Please define "early summer" in the Abstract and Section 1 (or 2). Does it refer to May and June (assumed from L104)? However, Meiyu rainfall occurs in June and July.

     Response: Early summer is defined as June–July in this study. We explained the definition of early summer in the abstract and the introduction: "early summer (June-July)." (Page 1 Line 11; Page 3 Line 86)

  2. It is notable that there are many extremely heavy Meiyu cases in the EQBO years (e.g., 1992, 1998, 2016, 2020). No clear reason has been given for examining only these two cases.

     Response: We revised the selection for the cases:

- "Three cases are selected to further verify the possible impact of the MJO phases 8-1 on eastern China heavy rainfall during EQBO. The total days of the MJO phases 8-1 exceed 20% of all days in June-July for those three years (1996: 21%; 2016: 36%; 2020: 54%)." (Page 14 Lines 300-302)

3.  In 2020, extreme Meiyu rainfall was associated with persistent phases 1-2 of the MJO (Zhang et al. 2021; Liang et al. 2021), which correspond to anomalous convection in the Indian Ocean, rather than phases 8-1 of the MJO and anomalous convection around the Maritime continent as this study emphasized.
    Response: Because the modulation effect of the QBO is more evident for MJO MJO phases 8-1 and 4-5 in our study, we emphasized on those MJO phases. To well address your concern, we also provide the composite for MJO phases 1-2 in Figure S16. We added a sentence:

- "Recent studies have identified that the MJO phases 1 and 2 have a significant impact on the 2020 extreme Meiyu rainfall (Liang et al., 2021; Zhang et al., 2021). The composite for MJO phases 1-2 is shown in Figure S16, and similar QBO modulation is also observed." (Page 15 Lines 314-316)

4.  Wang et al. (2022) examined the intraseasonal variation of the Meiyu rainbelt and its relationship to the MJO phase transit (L266-267), which has little relevance to our topic.
    Response: We removed this reference. (Page 14 Lines 312-314)

5.  Is Figure 7a the same as Fig. 2a? If yes, there is no need to show it.
    Response: Removed. (Page 7)

6.  Does the drought during phase 4-5 of MJO under WQBO contribute to the hot and dry conditions over eastern China in 2022?
    Response: Not really. The early summer in 2022 is in the EQBO phase.

**References:**

- Martin, Z., Son, SW., Butler, A. et al. The influence of the quasi-biennial oscillation on the Madden–Julian oscillation. Nat Rev Earth Environ 2, 477–489 (2021). https://doi.org/10.1038/s43017-021-00173-9

- Yoo, C., & Son, S. W. (2016). Modulation of the boreal wintertime Madden-Julian oscillation by the stratospheric quasi-biennial oscillation. Geophysical Research Letters, 43, 1392–1398. https://doi.org/10.1002/2016GL067762

- Wang, S., Tippett, M. K., Sobel, A. H., Martin, Z., & Vitart, F. (2019). Impact of the QBO on prediction and predictability of the MJO convection. Journal of Geophysical Research: Atmospheres, 124, 11766– 11782. https://doi.org/10.1029/2019JD030575

- Zhang, W., Huang, Z., Jiang, F., Stuecker, M. F., Chen, G., & Jin, F.-F. (2021). Exceptionally persistent Madden-Julian Oscillation activity contributes to the extreme 2020 East Asian summer monsoon rainfall. Geophysical Research Letters, 48, e2020GL091588. https://doi.org/10.1029/2020GL091588

- Liang, P., Hu, ZZ., Ding, Y. et al. The Extreme Mei-yu Season in 2020: Role of the Madden-Julian Oscillation and the Cooperative Influence of the Pacific and Indian Oceans. Adv. Atmos. Sci. 38, 2040–2054 (2021). https://doi.org/10.1007/s00376-021-1078-y

**Response to Reviewer # 2**

This paper explores the influence of the QBO and MJO on precipitation over Eastern China during May-June for the years 1979-2021. Statistically-significant results include enhanced rainfall during QBO E and MJO phases 8-1 and diminished rainfall during QBO W and MJO phases 4-5. Anomaly fields of upper tropospheric divergence, OLR, geopotential height, rainfall, lower tropospheric horizontal winds and moisture convergence, and meridional circulation in the plane for 110-120E support their diagnosis: During MJO 8-1, ITCZ convection near 10N is suppressed, and a region of anomalous 850 hPa high occurs near 20N, with one center over the Philippines which acts to transport more moisture into Eastern China from the south, and this is largest during QBO E. During MJO 4-5, ITCZ convection near 10N is enhanced and a region of anomalous 850 hPa low occurs near 20N, with the cyclonic anomaly over the Philipines bringing in cooler, dryer air over Eastern China, and this effect is largest during QBOW.
Response: Thank you for your positive comments.

The analysis methods and description of results are logical and clear. I have relatively minor suggestions for improving the figures. One point of diagnosis is that the two case studies of high rainfall seem to be in the near-neutral phase of the QBO instead of easterly at 50 hPa. This paper makes a useful contribution toward understanding the mechanisms of rainfall climate anomalies over East Asia. I recommend publication with minor revision and include some comments and suggestions below.

1. It might be good to add "and the Madden-Julian Oscillation" at the end of the title.
   Response: We added "and the Madden-Julian Oscillation" for the title.

2. line 10: perhaps specify "early summer rainfall (May – June) in eastern China"
   Response: All the analysis is for June-July rather than May-June. (Page 1 Line 11; Page 3 Line 85)

3. lines 49-51: For the polar stratosphere route, perhaps cite Holton and Tan 1980; for the tropical connection route, perhaps cite Collimore et al. 2003. Haynes et al. 2022 and Hitchman et al. 2022 may also be useful. (These are only suggestions.)
   Response: We added these references. (Page 2 Lines 52-54)

4. l54: add (PER)
   Response: Added. (Page 2 Line 57)

5. l107: maybe the second "0.5" in this line should be "0.7"
   Response: Corrected. (Page 4 Line 118)

6. I just want to be sure - is all of the analysis for May – June, including Fig. 7? Maybe state that in section 2.
   Response: All the analysis focuses on June-July rather than May-June. (Page 3 Line 85)

7. For the figures it would be helpful to

   a) reduce the white space and expand each panel to see details more easily,
   Response: We reduced the white space and expanded each panel in our figures.

   b) perhaps emphasize the country outline more clearly – I can see it on a computer screen but on a paper version it is hard to see – maybe outline the area of interest in China on each panel so that the reader can quickly see where it is relative to general circulation features
   Response: We emphasized the country outline more clearly in our figures this time.

   c) Consider if some figures can be made more uniform for latitude – longitude area shown, or in lieu of that, perhaps include a sentence or two in section 2 describing the need to look at different latitude ranges for different quantities. Highlighting the area of interest will help in going back and forth between figures with different domains.
   Response: The latitude and longitude are not uniform because the anomalous centers are different for different variables.
   - "It should be noted that different latitude ranges are chosen for different quantities to better show the key results." (Page 4 Lines 100-101)

8. Fig. 1 caption: perhaps state that "dashed lines show convergence anomalies"
   Response: Added. (Page 5 Line 133)

9. line 149, wave train: Does this refer to a pattern in the meridional plane?
   Response: The wave train means the deep convection in tropics can excite a pattern of Rossby wave train, which can be confirmed in both meridional plane (Figure 5) and 850 hPa geopotential height anomalies (Figure 3).

10. lines 143-145: Is it important that there are multiple centers? If it were a uniform east-west high pressure anomaly at 850 hPa, then maybe it would be harder to advect moist air into eastern China, but if the Philippine center is amplified, it can. If that is true you could further emphasize this aspect. The Philippine center seems to be important for both droughts and floods.
    Response: This sentence describes the intensification of the OLR anomalies and multiple centers during MJO phase 4-5 configured with EQBO, which do not correspond to intensified OLR anomalies. (Page 8, Lines 195-206; Page 9, Lines 216-222)

11. l163: Maybe this should read "4-5" instead of "8-1"?
    Response: Corrected. (Page 7 Line 183)

12. Fig. 5 caption: please state that green is upward motion to avoid any confusion.
    Response: The green is actually downward motion. We added "brown denotes upward motion" in the figure 5 caption.

13. Fig. 6e: this drying aspect for QBO W and MJO 4-5 is hard to see. Maybe if eastern China is highlighted in the panel it would be easier.
    Response: We added a highlight in the key zone in Figure 6 as you suggested.

14. Section 6: This is interesting! I looked up the QBO phase at 50 hPa and found that during May and June of 2020 there were weak easterlies (near zero), while in May and June of 1996 there were mild QBO W at 50 hPa (near zero). These transition phases are understudied and may have a stronger relationship that either QBO E or W with rainfall in China.
    Response: We use the anomalies of zonal mean zonal winds at 50-hPa to define the QBO phases considering the asymmetry of QBO westerlies and easterlies. The 2020 and 1996 early summers are both EQBO based on the 0.7 threshold in this study.

    - "The QBO index is defined as the anomalies of zonal mean zonal winds in the deep tropics (10°S-10°N) at 50 hPa." (Page 4 Lines 113-114)

**Response to Reviewer # 3**

TITLE: Modulation of the Intraseasonal Variability of Early Summer Precipitation in Eastern China by the Quasi-Biennial Oscillation

This manuscript examines the possible impact of MJO on early summer rainfall in eastern China and its modulation by QBO. Heavy rainfall in the boreal summer season is a topic of great importance for the region of East Asia due to its significant social impacts.

Response: Thank you for your positive comments.

The results presented by the author are very interesting and should eventually be published. However, I believe that the questions and concerns discussed below need to be addressed and the proposed revision work carried out.

Major Comments:

Comment 1:

Although this manuscript deals with the Asian monsoon region during the boreal summer, it does not mention boreal summer intra-seasonal oscillation (BSISO), the predominant mode in the region, at all. Kikuchi et al. (2012, DOI 10.1007/s00382-011-1159-1) showed that the amplitude of the MJO weakens during the boreal summer season, while the amplitude of the BSISO increases (see Fig. 6). They also showed that MJO and BSISO amplitudes are unlikely to strengthen simultaneously (see Fig. 4). The author presented composite maps based on MJO phase and amplitude, which are well similar to the characteristics of convective activity and circulation fields during BSISO dominance (see Fig. 8). Composite figures based on BSISO phase and amplitude should be constructed to see if the relationship with precipitation in East Asia can be seen more clearly.

Response: We showed the OLR, precipitation and circulation anomalies based on BSISO1 and BSISO2 phases in the supplementary this time (Figures S2-S13; BSISO1: Figures S2-S7; BSISO2: Figures S8-S13). However, the composite BSISO phases are not so evidently modulated by the QBO, although the OLR anomalies in BSISO1 and BSISO2 phases are also organized. The modulation of QBO on BSISO-related convection and precipitation is very similar to that for MJO. However, relation between the MJO and the WPH/SAH is more physically meaningful. In contrast, the SAH has little variation between QBO phases for BSISO. The WPH in BSISO1 phases 8-1 does not appear in WNP, which is not a typical pattern during Meiyu-Baiu period and fails to correspond to the persistent extreme rainfall event in June-July.

To well address your concern, we added a discussion about the BSISO-QBO link in the final section "summary and discussion":

"The tropical intraseasonal oscillation (ISO) has different propagation patterns between boreal summer and winter. The Boreal Summer ISO (BSISO) mode with prominent northward propagation and large variability in off-equatorial monsoon trough regions

is the prominent mode in boreal summer (Kikuchi et al., 2012). Although the amplitude of the MJO weakens during the boreal summer season, QBO is identified to affect the MJO-related convection and circulation in boreal summer based on statistical analysis in this research. Besides, the modulation of QBO on the BSISO-related convection and circulation is also inspected. The QBO has a consistent influence on the BSISO-related convection and circulation with MJO in boreal summer (Figures S2-S13; BSISO1: Figures S2-S7; BSISO2: Figures S8-S13) using the phase division by Lee et al. (2013). The modulation of QBO on BSISO-related convection and precipitation is similar to that for MJO phases. However, the SAH is not clearly modulated by the QBO phases for both BSISO1 and BSISO2. The WPH in BSISO1 phases 8 and 1 does not appear in WNP, which is inconsistent with the typical Meiyu circulation pattern." (Page 18 Lines 386-395)

Comment 2:

The author uses ERA5, but questions the use of NCEP-NCAR only for the geopotential height field and specific humidity. It is understandable if the author is comparing ERA5 and NCEP-NCAR, but further explanation is needed to convince the reader about using a different data set only for certain elements.
Response: The composite results for both ERA5 and NCEP-NCAR are provided in this study. Most of composites for NCEP-NCAR are shown in supplementary this time. The composites for ERA5 and NCEP-NCAR are very similar.

Minor Comments:

Comment 1: Page 1 L30-31 I agree that the MJO affects mid-latitude circulation in the Northern Hemisphere winter. However, I do not believe that it can be applied in the same way to the Northern Hemisphere summer season. This is because the basic fields are very different between winter and summer.
Response: We removed this explanation of mechanism for effect of the MJO on the Northern Hemisphere summer season. (Lines 30-31)

Comment 2: Page 2 L32: Alexander et al. (2018) does not discuss teleconnection to mid-tropospheric latitudes, so there is no need to refer to it?
Response: We removed this reference. (Page 2, Line 33)

Comment 3: Page 2 L35-36: Barnes et al. (2019) discuss the relationship between the NAO and polar vortex and the MJO, while Kang and Tziperman (2018) discuss the relationship between the MJO and the SSW and mid-latitude jet However, both do not directly discuss the annular mode.
Response: We replaced "northern annular mode" with "NAO". (Page 2 Line 36)

Comment 4: Page 2 L44: Wang et al. (2013) examined the impact of the MJO on the South China Sea during the boreal summer and did not discuss the relationship between the QBO and the MJO.

Response: Corrected. (Page 2 Line 45)

Comment 5: Page 2 L50: Garfinkel and Hartmann (2011) use a dry model to simulate jet variability in the upper troposphere and do not discuss tropical convective variability.
Response: We removed this reference. (Page 2 Line 53)

Comment 6: Page 2 L56: Does PER stand for persistent extreme rainfall?
Response: Yes. Corrected. (Page 2 Line 58)

Comment 7: Page 3 L90-91: I don't understand why you are using the geopotential altitude and specific humidity from the NCEP-NCAR reanalysis and not ERA5, I think you should use the ERA5 ones.
Response: We considered your suggestion.

Comment 8: Page 4 L99,L101: Shouldn't these formulae simply be $RMM1^2 + RMM2^2$ ?
Response: We checked and this formular is correct. (Page 4 Line 110; Page 4 Line 112)

Comment 9: Page 6 L136: As only a statistical relationship is shown here, I consider it more appropriate to refer to "statistical relationship between QBO phases and MJO-related rainfall anomalies" instead of "Modulation of ….".
Response: Corrected. (Page 6 Line 150)

Comment 10: Page 6 L143: Rao et al. (2023) makes comparisons between climate models of convective activity in the tropics, but does not specifically mention stability.
Response: We removed this reference. (Page 7 L158)

Comment 11: Page 7 L153: Does the Eastern Tropical Indian Ocean refer to the Bay of Bengal? If so, wouldn't it be clearer to describe it as the Bay of Bengal, since convection is suppressed in the eastern equatorial Indian Ocean?
Response: Corrected. (Page 7 Line 168)

Comment 12: Page 7 L164: "MJO phases 8-1" -> "MJO phases 4-5"?
Response: Corrected. (Page 7 Line 183)

Comment 13: Page 7 L170: "As an anomalous cyclone develops" "As an anomalous anti-cyclone (high) develops"?
Response: Corrected. (Page 8 Line 196)

Comment 14: Page 8 L190: "tropical Indian Ocean" -> "From the Arabian Sea to the Bay of Bengal" ?
Response: Corrected. (Page 9 Line 218)

Comment 15: Page 9 L214: "the Maritime Continent" -> "the South China Sea"?
Response: Corrected. (Page 11 Line 248)

Comment 16: Page 10 L236: How was the VIMF calculated? If it was calculated using NCEP-NCAR specific humidity and ERA5 winds, could inconsistencies arise? At first glance, the distribution of VIMF divergence deviations and VIMF deviations in Figure 6f does not seem to be consistent.

Response: Both reanalyses are calculated. ERA5 is shown in the supplementary.

"The VIMF is the thickness-weighted sum of the product for humidity and horizontal wind." (Page 13 Lines 272-273)

Comment 17: Page 13 L311-312: Rao et al. (2020a, 2023) do not specifically mention summer rainfall.

Response: We cite for the second half sentence (Page 16 Lines 364)

Comment 18: Page 14 L332-333: The results of the study considering ENSO are not presented anywhere in the text, but are very interesting and are strongly recommended to be shared in the supplement.

Response: We provided the relavent results in the supplementary.

- "The pure composite rainfall anomalies with the ENSO signals removed are shown in Figure S1 to examine the possible interference of ENSO with the composite results. The general patterns for the OLR composite are nearly unchanged (Figure 1 vs. Figure S1), although the amplitude of anomalous signals is enhanced around the south coast of China after the ENSO signals are removed. The influences of ENSO on the composite EQBO-WQBO difference are also very weak (Figure S1g and S1h)." (Page 7 Lines 189-193)

---

## Author Response (AR2)

**Cover letter**

Dear Prof. Peter Haynes,

Thank you for your letter and for the reviewers' comments concerning our manuscript. Those comments are all valuable and very helpful for revising and improving our paper, as well as the important guiding significance to our research. We have studied the comments carefully and made corrections which we hope meet with approval. The main corrections in the paper and the responses to the reviewer's comments are attached. We hope our revisions have well addressed every concern from our reviewers.

Yours,
Sincerely,

Jian Rao
raojian@nuist.edu.cn

**Response to Editor**

The referees all agree that your paper is close to being ready for publication. Referees 2 and 3 are requesting further minor changes. Please consider their comments carefully and also look at the whole paper carefully again to consider whether modest changes might improve clarity. I have made various comments below about the referee comments and about some other points in the paper.

Response: Thank you for your positive comments. We considered the comments from you and referees 2 and 3.

Please provided a revised version of the paper including responses to the referees' comments and to my own comments below. I hope that at the next stage I will be able to accept the paper for publication without further consultation with the referees, but this will depend on the changes that you have made and the clarity of your responses.

Response: Thank you for your kindness.

**Response to Reviewer # 2**

Referee 2: 'The only further point that the committee would like to discuss is that the figures in the text should be unified with those of ERA5. ERA5 should be prioritized because it is of higher quality than the NCEP/NCAR reanalysis in the tropics. Figures generated by the NCEP/NCAR reanalysis should be included in the supplement for reference purposes. Also, changing the data used for different elements without explanation may give the reader the impression that the figures in the paper are intentional.'

I think that 'the committee' should be interpreted as 'the referee'. The point being made here is that you use NCEP/NCAR for some purposes and ERA5 for others. This seems a valid point to me. Given that ERA5 has now been available for some time, why are you not using it for all quantities. If there is a concrete reason why you are not using it -- e.g. you have tried using ERA5 for some fields and the results you find are not as clear as when you use NCEP/NCAR then that should be mentioned explicitly. My recommendation is, following the referee comments, that you use ERA5 as the primary data source and that it you do not then you provide clear explanation for that.

Response: Thank you for the suggestion. We revised the manuscript with the ERA5 data replacing the NCEP/NCAR. The results based on the NCEP/NCAR are moved to the supplementary materials. Specific revisions are listed as follows.

- We changed the figures 1 and 6 with the ERA5 data and put the figures derived from NCEP/NCAR reanalysis into supplementary S17 and S21.
- Besides, we added the NCEP/NCAR version of Figure 5 as Figure S20. The figures of ERA5 and NCEP/NCAR reanalysis showed quite consistent results.
- We also changed the data description because we only use the ERA5 data in the paper this time. (L88-101)

**Response to Reviewer # 3**

Referee 3: I have a concern about descriptions of mechanisms and phenomena involved. The discussion is rather rambling, with many different things thrown into the stew. I recommend that this work should be published, but I feel that the authors need to try harder to streamline the discussion of physical mechanisms, to clarify and emphasize the primary processes that they think explain their result.'

Again I can understand why the referee has made this comment. You list 'main findings' in Section 7, but there are 7 main findings and several of them are quite complicated. Please consider whether your paper will be more effective if your choice of 'main findings' is more restricted and if those that you choose are genuinely robust and important.

Response: Thank you for the suggestion. We condensed the main findings this time Please refer to the last section (Page 16).

In considering the above I noted the following detailed points -- there may be others.
Section 2: I found it quite difficult to find a definition of 'early summer' and how that affects your use of data. In the abstract you define early summer as June-July. Does that mean that you consider MJO indices etc ONLY in June-July. How is this implemented exactly -- for example if data is low-pass filtered in time then 'June-July' may be influenced by other months. Please be clear about your methods. The issue is slight confused by the fact that you say that the QBO phase is defined by fields in May-June.

Response: Thank you for the suggestion. We agree that the RMM index in June-July from Bureau of Meteorology Australia (BoM) indeed applied the band pass filter (20-100 day) to original data (OLR, U850 and U200) anomalies before applying a MV-EOF analysis. However, the projected timeseries of the MV-EOFs are not filtered and are raw data. The timeseries in June and July are not affected by other months.

We made several revisions this time as follows:

- "The raw Real-time Multivariate MJO (RMM) index (Wheeler and Hendon, 2004) is used to define the MJO phase." (L101)
- "Note that the early summer is defined as June-July to focus on the influence of the QBO and MJO during the Meiyu period." (Line 119-120).
- "The index in May-June (May–July means show similar results) is used to select the QBO phases…" (L113)

L309: 'and its modulation by the MJO' -- should this be 'and its modulation by the QBO'?
Response: Revised (L315).